# Genetic Analyses of *Saprolegnia* Strains Isolated from Salmonid Fish of Different Geographic Origin Document the Connection between Pathogenicity and Molecular Diversity

**DOI:** 10.3390/jof7090713

**Published:** 2021-08-30

**Authors:** Abdelhameed Elameen, Svein Stueland, Ralf Kristensen, Rosa F. Fristad, Trude Vrålstad, Ida Skaar

**Affiliations:** 1Division of Biotechnology and Plant Health, Norwegian Institute of Bioeconomy Research (NIBIO), N-1431 Ås, Norway; 2Norwegian Veterinary Institute, Pb 64, N-1431 Ås, Norway; postmottak@vetinst.no (S.S.); ralf.kristensen@vetinst.no (R.K.); rosa.fristad@vetinst.no (R.F.F.); trude.vralstad@vetinst.no (T.V.); ida.skaar@vetinst.no (I.S.)

**Keywords:** AFLP, fingerprinting, genetic diversity, ITS, origin, salmon

## Abstract

*Saprolegnia parasitica* is recognized as one of the most important oomycetes pests of salmon and trout species. The amplified fragment length polymorphism (AFLP) and method sequence data of the internal transcribed spacer (ITS) were used to study the genetic diversity and relationships of *Saprolegnia* spp. collected from Canada, Chile, Japan, Norway and Scotland. AFLP analysis of 37 *Saprolegnia* spp. isolates using six primer combinations gave a total of 163 clear polymorphic bands. Bayesian cluster analysis using genetic similarity divided the isolates into three main groups, suggesting that there are genetic relationships among the isolates. The unweighted pair group method with arithmetic mean (UPGMA) and principal coordinate analysis (PCO) confirmed the pattern of the cluster analyses. ITS analyses of 48 *Saprolegnia* sequences resulted in five well-defined clades. Analysis of molecular variance (AMOVA) revealed greater variation within countries (91.01%) than among countries (8.99%). We were able to distinguish the *Saprolegnia* isolates according to their species, ability to produce oogonia with and without long spines on the cysts and their ability to or not to cause mortality in salmonids. AFLP markers and ITS sequencing data obtained in the study, were found to be an efficient tool to characterize the genetic diversity and relationships of *Saprolegnia* spp. The comparison of AFLP analysis and ITS sequence data using the Mantel test showed a very high and significant correlation (*r*^2^ = 0.8317).

## 1. Introduction

Saprolegniasis causes great damage and infection in fish in aquaculture and fish farms [1], and *Saprolegnia parasitica* is recognized as one of the most important oomycetes pests of salmon and trout species in Scandinavia, Chile, Japan, Canada and Scotland [2]. Thus, it causes losses of tens of millions of dollars in aquaculture businesses worldwide [1]. *Saprolegnia* spp. are generally termed “watermolds” and share common features with both fungi and algae [3]. All fish and ova in fresh water can possibly be infected by *Saprolegnia* spp., and the disease is termed saprolegniasis. Infected fish are easily recognized by the cotton-like white to greyish patches on the skin and gills [4]. During the last few decades there has been an increased focus on saprolegniasis in salmonid fish and a number of *Saprolegnia* outbreaks, and attempts to characterize *Saprolegnia* isolates have been reported [5,6]. Differences in pathogenicity have been proved between strains of a *Saprolegnia* species even within the same taxonomic grouping [7,8,9,10].

Traditionally, taxonomic characterization of *Saprolegnia* spp. has been based upon morphological and physiological characteristics [8,11], but in recent years, molecular methods have become useful tools when describing phylogeny, taxonomy and epidemiology [12,13,14,15,16]. Epidemiological studies of *Saprolegnia* spp. are particularly useful for identifying sources of infection, characterizing disease spread and improving disease management. However, molecular studies of *Saprolegnia* isolates are still limited [6], and an improved knowledge is important in order to reduce *Saprolegnia* outbreaks in the future.

Our aim was to investigate the genetic diversity and relationships of *Saprolegnia* spp. isolates collected from Canada, Chile, Japan, Norway and Scotland. Our hypotheses were that genetic diversity within and among these countries expresses broader genetic structures or, alternatively, that gene flow is high across all these countries, due to the trade and exchange of breeding materials. Furthermore, we tested if molecular markers (amplified fragment length polymorphism (AFLP) and sequence data of the internal transcribed spacer (ITS)) could distinguish the *Saprolegnia* isolates according to their species, their ability to produce oogonia with and without long spines on the cysts, and their ability to or not to cause mortality in salmonids. To achieve these goals, we investigated *Saprolegnia* spp. isolates (Table 1) using AFLP and ITS markers. We believe our study may provide novel insights into the genetics and biology of *Saprolegnia* spp., as well as indirect knowledge for disease management.

## 2. Materials and Methods

### 2.1. Saprolegnia Material

Thirty-seven *Saprolegnia* spp. strains were collected from salmonids in Canada (5), Chile (6), Norway (14), Scotland (9) and Japan (3) were investigated in this study (Table 1). The sampling procedure and all morphological, physiological and pathogenic characteristics were described by Stueland et al. [10]. Most strains were isolated from cultured Atlantic salmon (Salmo salar L.), ova, fry or brood stock suffering from saprolegniasis. However, the three *Saprolegnia* species from Japan were isolated from cultured sockeye salmon (Oncorhynchus nerka) and Coho salmon (Oncorhynchus kisutch) [7,9,17,18,19]. One of the Norwegian *Saprolegnia* strains was isolated from brown trout (Salmo trutta) and one strain from common whitefish (*Coregonus lavaretus*) (Table 1). Twenty-five of the strains have previously been analyzed with respect to their morphological and physiological characteristics [10], and 9 of these strains have also been previously characterized with regard to their pathogenicity to Atlantic salmon [10].

### 2.2. Amplified Fragment Length Polymorphism (AFLP) Analysis

DNA extraction of mycelia was performed by use of the Puregene Gentra DNA Isolation Kit (Gentra Systems, Minneapolis, MN, USA) according to the manufacturers’ description. The DNA pellets were dissolved in 50 μL of a TE buffer (pH 7.5) and stored at −20 °C. AFLP analysis was performed as previously described [20]. A total of 300 ng of genomic DNA was double-digested with *EcoRI* and the *MseI* isoschizomer *Tru1I*. Following ligation of the restriction fragments to the adaptors, pre-amplification PCR was carried out with non-selective primers in a total volume of 25 µL containing 5 µL of fivefold diluted ligation product. Ten primer combinations with two selective nucleotides were chosen for a pre-screening of eight randomly chosen isolates. Six highly polymorphic primer pairs were selected and used to generate AFLP fragments of all 37 isolates (Table 2). After the selective amplification using *EcoRI* primers end-labeled with γ-P^33^-ATP (3000 Ci/mmol^−1^), the PCR products were separated on 5% polyacrylamide gels. All primer combinations were repeated twice.

### 2.3. Data Analysis of the AFLP

Data were recorded manually, and only clear polymorphic bands were scored for presence (1) or absence (0). The genetic similarity (GS) was estimated using the Dice coefficient, calculated as GS_xy_ = 2a/(2a + b + c), where a is the number of bands present in both isolates, b is the number of bands present only in isolate x and c is the number of bands present only in isolate y [21]. The genetic similarity among the clones, based on the presence or absence of amplified fragments, was also calculated by Jaccard coefficients [22]. Both analyses resulted in the same clusters, and only the results obtained by the Dice coefficient are presented.

We measured the percentage of polymorphic bands in four countries, calculated by dividing the number of polymorphic bands at the country level by the total number of bands scored. The estimates were based on the isolates from four of the countries. Isolates from Japan were excluded from the analyses, since this country was represented by fewer than five isolates.

The matrix of similarity data was analyzed using the unweighted pair group method with arithmetic mean (UPGMA), as suggested by Sneath and Sokal (1973) [23]. UPGMA clustering was also carried out for all the isolates according to their country of origin, their ability to produce oogonia with and without spines, and their pathogenicity. We performed principal coordinate (PCO) analysis to classify and detect the structure of the relationships between the isolates with different countries of origin, differing ability to produce oogonia with and without spines, and differing pathogenicity. Statistical analyses and construction of dendrograms were performed using NTSYS-pc software version 2.1 [24].

Analyses of molecular variance (AMOVA) [25] was carried out using Arlequin software, version 2.000 [26]. The analysis was estimated according to the country of origin. Isolates from Japan were eliminated from the study because they were represented by fewer than five individuals. The genetic distances among *Saprolegnia* spp. isolates from Canada, Chile, Norway and Scotland were calculated using the pairwise genetic distance method [27]. The mean F_ST_ was estimated in order to study genetic differentiation between countries. The significance of the *F*_ST_ values was tested by 1000 permutations. These analyses were performed using Arlequin software, version 2.000 [26]. Gene flow was estimated by assuming Nm = (1/_FST_ − 1)/4 [28].

### 2.4. Sequencing of the ITS

The internal transcribed spacer 2 (ITS2) part of the nuclear ribosomal DNA was amplified with the primer pair ITS2/ITS4 according to White, Bruns, Lee and Taylor (1990) on a PTC-225 (Peltier Thermal Cycler, MJ Research, Waltham, MA, USA). PCR amplicons were purified with ExoSap IT (GE healthcare, Buckinghamshire, UK) according to the manufacturer’s procedure and visualized on standard agarose gel to ensure the presence of single-band products. Both strands of the PCR amplicons were sequenced with the PCR primers using DYEnamic ET dye terminator chemistry (Amersham Biosciences, Chicago, IL, USA), purified on AutoSeq96 (Amersham Biosciences) plates, diluted with 10 µL of MQ-water and subsequently analyzed on a MegaBace 1000 (Amersham Biosciences). Sequences were analyzed in Vector NTI Advanced 11 (Invitrogen, Waltham, MA, USA) and assembled in BioEdit 7.0.9.0 [29].

### 2.5. Phylogenetic Analyses

All new sequences are available from the EMBL/GenBank sequence database under accessions FN186015–FN186051 (Table 1). In addition, we downloaded and included the following sequence accessions: EU551153 (S. *salmonis*), AM228727 (*S. parasitica*), AM228814 (*S. hypogyna*), EF126339 (*S. mixta*), EU124763 (*S. ferax*), EU152130 (S. sp. *nuchiae*), AY270032 (*S. longicaulis*), EU124765 (*S. diclina*), AM228818 (*S. australis*), AM947036 (*S.* cf. *ferax*) and AY310502 (*Leptolegnia* sp.) based on similarity analyses against the EMBL/GenBank sequence database. The sequence alignment included a total of 48 isolates (46 *Saprolegnia* and 2 *Leptolegnia*), with ITS2 sequences 685–691 bp in length, representing at least 11 species and 35 haplotypes. Sequences were manually aligned; the data matrices are available from TreeBASE (Table 1). ITS sequences were selected from the correct data base published by Matthews et al. (2021) [30]. Phylogenetic analyses of the ITS were performed using maximum parsimony, using a heuristic search (2000 random replicates) in PAUP* v4.0b10 [31]. Gaps were treated as missing, and isolate N66, *Leptolegnia* sp., was selected as an outgroup based on previous phylogenetic studies of *Saprolegnia* [6]. To evaluate the support for the observed branching topologies for maximum parsimony, we performed bootstrap analysis [32], with 2000 bootstrap replicates. Bootstrapping was carried out with a heuristic algorithm and 2000 random additions of sequences per bootstrap replicate.

### 2.6. Cluster Analyses

The genetic structure of the *Saprolegnia* isolates were also investigated using the model-based Bayesian clustering approach of genetic admixture analysis (Structure 2.3.4 software) [33]. Simulations were performed with a dataset from K = 1 to K = 7. The method developed by Evanno et al. (2005) [34] was used to identify the number of genetically homogeneous clusters (K). We used a burn-in period of 100,000 runs and 500,000 MCMC runs to compute the probability of the data for estimating K. Among the seven independent runs, the one with the highest Ln Pr (X|K) value (log probability) was chosen and represented as bar plots. Bar plots of likelihoods and ΔK values were made with STRUCTURE HARVESTER [35].

## 3. Results

AFLP analyses of all 37 isolates using six primer combinations resulted in a total of 163 clear polymorphic bands (Table 2). The number of polymorphic bands per primer combination ranged from 23 to 31 bands, with an average of 27.17 polymorphic bands per primer pair. With these six primer combinations, it was possible to uniquely distinguish among all 37 isolates. Genetic similarity for *Saprolegnia* isolates using Dice coefficients based on the AFLP data ranged from 0.0 to 1.0, with a mean of 0.41. The highest genetic similarity (1.0) was obtained between two isolates, one from Norway and the other from Scotland, while the lowest genetic similarity value (0.0) was observed between two isolates from Norway.

The percentage of polymorphic bands (PPB) in each group (i.e., country) was high, ranging from 29.45% to 84.05%, with a mean value of 55.52%. The Norwegian isolates showed the highest PPB and the Canadian showed the lowest PPB (Table 3).

UPGMA clustering of the *Saprolegnia* isolates, based on their countries of origin, did not consistently reflect the geographic origin of the *Saprolegnia* isolates. Some *Saprolegnia* isolates from Norway clustered together with isolates from Scotland (Figure 1). However, based on UPGMA, the *Saprolegnia* isolates were completely clustered according to their ability to produce oogonia with and without long spines on the cysts (Figure 1). We detected 15 specific markers that differentiated isolates that had the ability to produce oogonia with long spines from those without spines. UPGMA analysis of the *Saprolegnia* isolates according to their pathogenicity resulted in three clusters. All pathogenic isolates clustered together in one group, whereas the non-pathogenic isolates clustered together in two groups (Figure 1).

The results of the PCO analyses supported the results of the UPGMA analysis. There was no distinguishable clustering pattern of *Saprolegnia* isolates from a certain country, but the isolates were completely grouped according to their ability to produce oogonia with spines or without spines, and according to their ability to cause mortality or not in salmonids (Figure 2).

In general, the genetic distances between the regions measured by pairwise differences were low (Table 4), the highest genetic distance (0.25198) was between Canada and Chile, and the lowest was between Norway and Scotland (0.04209).

The AMOVA analyses based on the geographic origin of the isolates showed that most of the total genetic variability, i.e., 91.01%, was attributed to the variance within countries, while the among-country variance component only accounted for 8.99% (Table 5). A low proportion of the observed genetic differentiation can be explained by the level of the F_ST_ value (0.089), and the average estimated mean of gene flow (Nm) between the countries was relatively high (Nm = 2.559).

As regards sequencing of the ITS, maximum parsimony analyses with a heuristic search gave the 30 most parsimonious trees (MPTs) with a length of 117 steps. Figure 3 shows a bootstrap tree where all bootstrap values above 50 are indicated on the representative branches. The phylogenetic analyses conducted on 48 partial ITS sequences of *Saprolegnia* resulted in five well-supported clades. The majority of strains, namely 26, formed a clade with *S. salmonis* and *S. parasitica* sequences retrieved from EMBL/GenBank. All isolates within this complex either produced cysts with long hairs (Table 1) or were not tested. A well supported sister clade of this *S. parasitica* complex consisted of two strains, and one of these was identified as *S. hypogyna*. The third clade consisted of strains identified as *S. ferax, S.mixta, S. longicaulis* and *S. sp. nuchiae*. Some of these different species have identical ITS haplotypes. The last two clades were well-supported and formed sister clades. Five isolates grouped with two strains previously identified as *S. diclina* and *S. australis*. The last two isolates grouped with one strain previously identified as *S.* cf. *ferax*. Strain S5 was received as a *Saprolegnia* sp. strains, but both the phylogenetic analysis and the EMBL/GenBank Blast search revealed that this strain clearly belongs to *Leptolegnia*. Based on the phylogenetic analyses, no geographic grouping could be observed.

Bayesian clustering analysis with STRUCTURE software assigned the 37 *Saprolegnia* strains to three different clusters. Structure analysis showed the maximum likelihood distribution L(K) of the real number of three groups (K = 3). This value was obtained using the value of ad hoc quantity (ΔK) rather than maximum likelihood value L(K), as described by Evanno et al. (2005) [32]. Structure analysis clustered the *Saprolegnia* isolates into three main clusters (Figure 4) as the UPGMA and PCO groupings.

## 4. Discussion

The present study provides, for the first time, a good indication that the different pathogenic isolates of *Saprolegnia* strains clustered at a molecular level. Thereby, we provide documentation that there is a connection between pathogenicity to Atlantic salmon and the molecular diversity of *Saprolegnia* strains. The pathogenic *Saprolegnia* strains included in this study that formed one genetic cluster also shared the same morphological characteristics, i.e., long hooked hairs on the secondary zoospore cysts. Several reports have tried to prove that the variability among *Saprolegnia* isolates is correlated to infectivity [7,8]. We have previously shown significant differences in pathogenicity among seven of the strains included in the present study, and concluded that initial growth rate of germinating cysts in pure water, together with the presence of long hooked hairs on the secondary cysts, may be an indicator of the pathogenicity of *Saprolegnia* strains to Atlantic salmon [10]. In this study, we have extended this also to include characterization at the genetic level, reassuring our initial findings. However, the number of strains tested in vivo in the present study was limited for animal welfare reasons as well as the high cost of performing in vivo pathogenicity testing. Thus, these findings should be followed by further investigations including in vivo studies.

### 4.1. Geographical Origin

The present study proved there is more genetic variation among *Saprolegnia* strains within each country than among the countries included in the study. UPGMA clustering, phylogenetic analyses, PCO and STRUCTURE analyses consistently reflected the geographic origin of the *Saprolegnia* isolates. Actually, the AMOVA showed that most of the total genetic variability was attributed to the variance within countries. These results suggest that *Saprolegnia* from the countries included in this study share much of the same genetic material, which was also supported by the average estimated mean of gene flow (Nm) among the countries, which was relatively high (Nm = 2.559). This is in contrast to Bangyeekhun et al. (2003) [13], who reported the genetic dissimilarity of pathogenic *Saprolegnia* from different geographic locations (Northern Europe, Southern Europe and USA) in their study. Factors to consider are the industrialized nature of aquaculture and the transfer of fish among the countries included in the present study. This is an important risk factor for pathogen transfer in general. All the *Saprolegnia* strains included in the AFLP analysis were sampled from Atlantic salmon-farming countries, with fish/eggs being originally exported from Norway. In this context, it is not surprising that there is more genetic variation among *Saprolegnia* strains within each country than among the countries included in the study. The high gene flow detected in the study may be due to the repeated *Saprolegnia* introduction events among these countries and a low level of sexual recombination over time. These results are in agreement with the result detected by Paul et al. (2018) [36].

### 4.2. Morphology

The long hairs on the zoospore cysts seem to be a characteristic dividing the *Saprolegnia* strains into different genetic clusters. Wiilloughby (1985) [37] stated that this characteristic of *Saprolegnia* strains isolated from fish is typical for *S. parasitica*. Beakes et al. (1994) [38] suggested that *Saprolegnia* strains isolated from infected fish that had the distinctive clusters of long spines on their secondary cysts fulfilled an updated species concept of *Saprolegnia parasitica* in function, if not in taxonomic context. However, the taxonomic status of *Saprolegnia* species isolated from fish has still not been resolved. Dieguez-Uribeondo et al. (2007) [6] stated that the species-level identification of parasitic isolates of *Saprolegnia* has, at best, proven problematic and, at worst, impossible. In the present study, all *Saprolegnia* strains with long hooked hairs on the secondary zoospore cysts grouped together into a single major cluster, suggesting that they form a separate taxon.

### 4.3. Phylogeny

The phylogenetic analyses of 48 ITS *Saprolegnia* sequences resulted in five well-defined clades with *Leptolegnia* as an outgroup. These well-defined clades are largely congruent with previous studies [6]. All strains of *S. parasitica* formed a well-defined clade consistent with Clade I in a previous study [6]. In the present study, the term *S. parasitica* has been used for all strains within this clade. All strains tested herein within the *S. parasitica* complex produced cysts with long hairs. The *S. hypogyna* strains formed a very well-supported sister clade (97%) and corresponded to Clade Ia in [6]. The AFLP results were, for the main part, congruent with these groupings, with the exception of *S. hypogyna*, which grouped within the *S. parasitica* complex. The third clade consisted of at least four *Saprolegnia* species. This highly supported clade (100%) corresponds to Clade II proposed by Dièguez-Uribeondo et al. (2007) [6] but contains at least an additional three species. The grouping of the last two clades was very well supported (bootstrap: 94%). One of these clades consists of a combination of *S. diclina* and *S. australis* strains. The other clade consists of three strains, one identified as *S.* cf. *ferax*. These isolates are well separated from *S. ferax* and are therefore most likely a different species. This grouping into two clades is not congruent with earlier studies [6], which separated these species into three clades, designated III, IV and V. They included a broader range of species and a higher number of isolates for each species than the present study, which may explain the few observed differences between the two studies.

Most of the species included in the present study have ITS haplotypes in EMBL/GenBank that are identical to those of other species of *Saprolegnia*. This illustrates a common problem with ITS and fungal and fungal-like organisms: the lack of resolution to distinguish closely related species. In other well-studied fungi of fungal-like genera, there has been a shift from ITS to other genetic markers (e.g., β-tubulin, translation elongation factor 1-α). To reveal the true phylogeny of the genus *Saprolegnia*, other genetic markers should be used.

## Figures and Tables

**Figure 1 jof-07-00713-f001:**
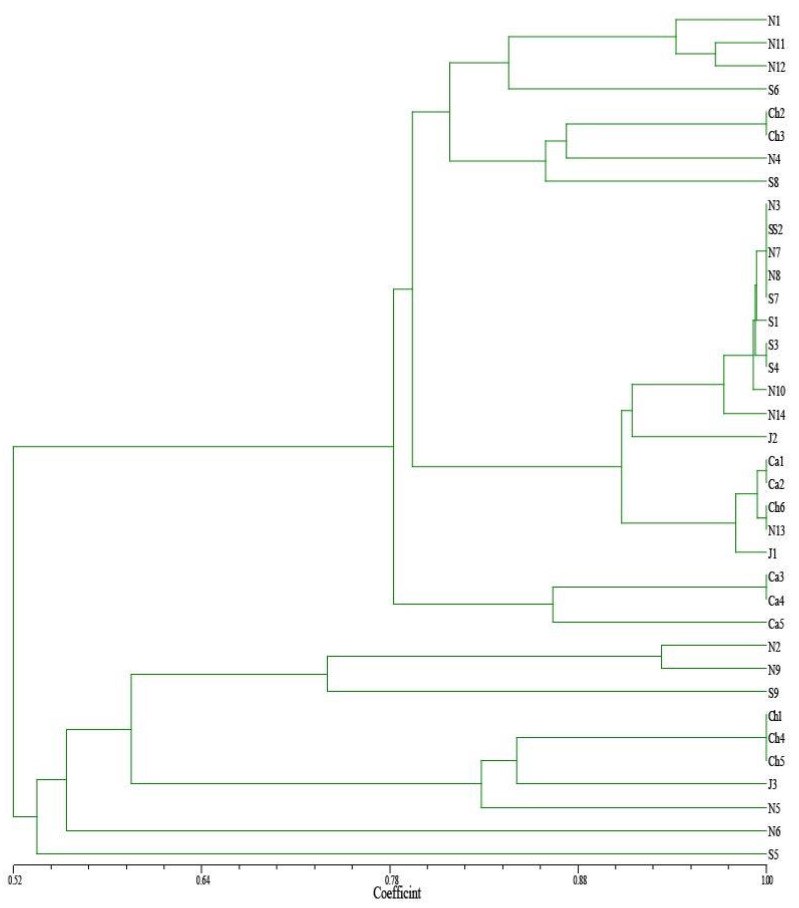
Dendrogram of isolates of *Saprolegnia* spp. as revealed by UPGMA cluster analysis of AFLP-based genetic similarity (Dice coefficient). The isolates are described by their geographic origin, as shown in Table 1.

**Figure 2 jof-07-00713-f002:**
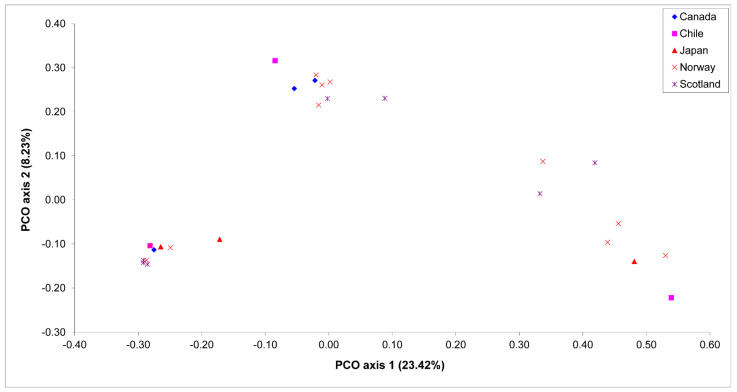
Score plot of principal coordinate analysis (PCO) of isolates of *Saprolegnia* spp. as revealed by using 163 polymorphic AFLP bands, collected from Canada, Chile, Japan, Norway and Scotland.

**Figure 3 jof-07-00713-f003:**
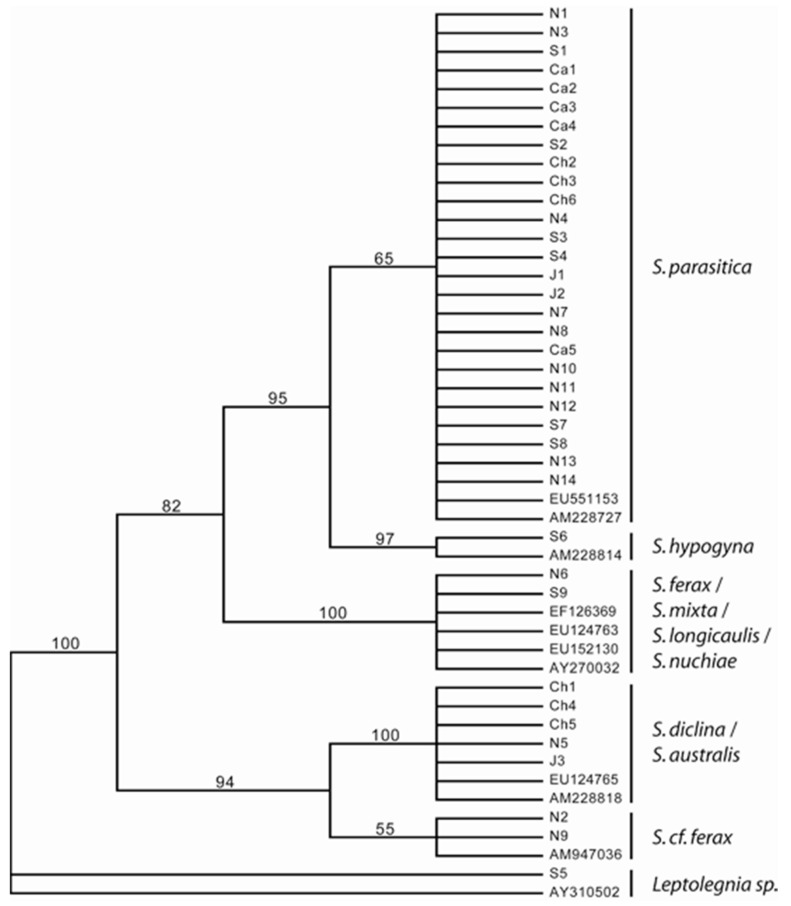
Phylogram of isolates of *Saprolegnia* spp. as revealed by sequencing of ITS region 2. The isolates are described by their geographic origin, their ability to produce long spines on cysts and their pathogenicity to salmon (Table 1).

**Figure 4 jof-07-00713-f004:**
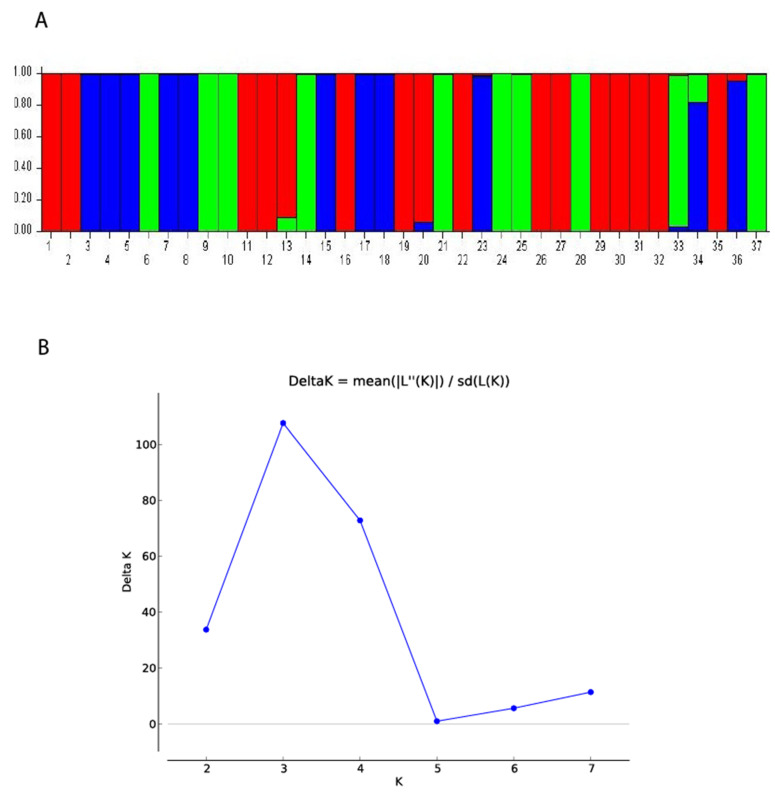
Structure analyses summary. (**A)** Bar plots showing the relationships for all 37 *Saprolegnia* strains in the study assessed on the basis of AFLP data; each strain is represented by a vertical line. The *Saprolegnia* isolates are sorted according to their ID number (Table 1). (**B**) The values for seven independent runs for K = 1–7 and DeltaK = mean (L’’(K)I)/s(L(K)).

**Table 1 jof-07-00713-t001:** The *Saprolegnia* isolates included in the study collected from Canada, Chile, Japan, Norway and Scotland.

Strain Id	Species	Strain Number *	EMBL Accession Number	Origin	Host	Cyst Coat **	Germ Rate ***	Path Test ****
N1	*S. parasitica*	VI 02388	FN186015	Norway	As parr	+	Slow	-
N2	*S.* cf. *ferax*	VI 02389	FN186016	Norway	AS parr	−	NT	NT
N3	*S. parasitica*	VI 02391	FN186017	Norway	As parr	+	Fast	++
N4	*S. parasitica*	VI 02750	FN186018	Norway	As parr	+	NT	NT
N5	*S. diclina*	VI 02753	FN186019	Norway	As parr	−	NT	NT
N6	*S. ferax*	VI 02756	FN186020	Norway	As parr	−	Slow	−
N7	*S. parasitica*	VI 02763	FN186021	Norway	Bt broodstock	+	NT	NT
N8	*S. parasitica*	VI 02770	FN186022	Norway	As parr	+	Fast	NT
N9	*S.* cf. *ferax*	VI 04808	FN186023	Norway	As fry	NT	NT	NT
N10	*S. parasitica*	VI 04810	FN186024	Norway	As parr	NT	NT	NT
N11	*S. parasitica*	VI 04811	FN186025	Norway	As parr	NT	NT	NT
N12	*S. parasitica*	VI 04812	FN186026	Norway	As parr	NT	NT	NT
N13	*S. parasitica*	VI 04817	FN186027	Norway	As broodstock	NT	NT	NT
N14	*S. parasitica*	VI 04819	FN186028	Norway	Cwf	NT	NT	NT
S1	*S. parasitica*	VI 02392	FN186029	Scotland	As parr	+	Fast	NT
S2	*S. parasitica*	VI 02736	FN186030	Scotland	As parr	+	Fast	++
S3	*S. parasitica*	VI 02757	FN186031	Scotland	As parr	+	Fast	NT
S4	*S. parasitica*	VI 02758	FN186032	Scotland	As parr	+	Fast	NT
S5	*Leptolegnia sp.*	VI 04813	FN186033	Scotland	As parr	NT	NT	NT
S6	*S. hypogyna*	VI 04814	FN186034	Scotland	As parr	NT	NT	NT
S7	*S. parasitica*	VI 04815	FN186035	Scotland	As parr	NT	NT	NT
S8	*S. parasitica*	VI 04816	FN186036	Scotland	As parr	NT	NT	NT
S9	*Saprolegnia sp.*	VI 04818	FN186037	Scotland	As parr	NT	NT	NT
Ca1	*S. parasitica*	VI 02393	FN186038	Canada	As parr	+	Slow	NT
Ca2	*S. parasitica*	VI 02394	FN186039	Canada	As parr	+	Fast	NT
Ca3	*S. parasitica*	VI 02395	FN186040	Canada	As parr	+	Medium	−
Ca4	*S. parasitica*	VI 02437	FN186041	Canada	As parr	+	NT	NT
Ca5	*S. parasitica*	VI 04809	FN186042	Canada	Ps	NT	NT	NT
Ch1	*S. diclina*	VI 02739	FN186043	Chile	As eggs	−	Slow	−
Ch2	*S. parasitica*	VI 02740	FN186044	Chile	As broodstock	+	Medium	−
Ch3	*S. parasitica*	VI 02741	FN186045	Chile	As eggs	+	Medium	NT
Ch4	*S. diclina*	VI 02744	FN186046	Chile	As eggs	NT	Slow	NT
Ch5	*S. diclina*	VI 02746	FN186047	Chile	Cs eggs	NT	NT	NT
Ch6	*S. parasitica*	VI 02748	FN186048	Chile	As eggs	+	Fast	NT
J1	*S. salmonis*	NJM 9851	FN186049	Japan	Ss	+	Fast	++
J2	*S. parasitica*	ATCC90213	FN186050	Japan	Cs parr	+	Fast	++
J3	*S. diclina*	ATCC90215	FN186051	Japan	Cs parr	−	Slow	−

***** Refers to the number in the strain collection of the National Veterinary Institute. Atlantic salmon (Salmo salar L.) (As), Ca (Canada), Ch (Chile), J (Japan), N (Norway) and S (Scotland). Coho salmon (Oncorhynchus kisutch) (Cs), sockeye salmon (Oncorhynchus nerka), (Ss), Pacific salmon (Oncorhynchus tshawytcha) (Ps), brown trout (Salmo trutta) (Bt) and common whitefish (Coregonus lavaretus) (Cwf). ** Long hairs (+) or no long hairs (−) on the secondary zoospore cyst. *** Germination rate (% cysts germinating) after incubation of Saprolegnia mycelium in sterilized tap water (STW) at 21 ± 1 °C for 20 hours. Slow = 0–10%; medium = 11–49% and Fast = 50–100% cysts germinating. **** High (++), medium (+) or low(−) mortality in Atlantic salmon exposed to Saprolegnia cysts (1 × 10^4^ cysts/L). NT = not tested.

**Table 2 jof-07-00713-t002:** Nucleotide sequences of the selective primers used for AFLP analyses and the number of polymorphic bands resulting from each primer combination.

Primer Combination	*EcoR*I Primer 5′ → 3′	*Mse*I Primer 5′ → 3′	Number of Polymorphic Bands
E12 × M17	GAC-TGC-GTA-CCA-ATT-CAC	GAT-GAG-TCC-TGA-GTA-ACG	27
E13 × M15	GAC-TGC-GTA-CCA-ATT-CAG	GAT-GAG-TCC-TGA-GTA-ACA	28
E11 × M16	GAC-TGC-GTA-CCA-ATT-CAA	GAT-GAG-TCC-TGA-GTA-ACC	23
E20 × M17	GAC-TGC-GTA-CCA-ATT-CGC	GAT-GAG-TCC-TGA-GTA-ACG	31
E21 × M17	GAC-TGC-GTA-CCA-ATT-CGG	GAT-GAG-TCC-TGA-GTA-ACG	30
E22 × M17	GAC-TGC-GTA-CCA-ATT-CGT	GAT-GAG-TCC-TGA-GTA-ACG	24

**Table 3 jof-07-00713-t003:** The number and percentage of polymorphic bands resulting from the AFLP analyses of *Saprolegnia* spp. in the study.

Country of Origin	No. Polymorphic Bands	% Polymorphic Bands
Canada	48	29.45
Chile	86	52.76
Norway	137	84.05
Scotland	91	55.83
Mean	90.5	55.52

**Table 4 jof-07-00713-t004:** The average genetic distances among *Saprolegnia* spp. calculated by the genetic distance method (Exoffier and Smouse 1994) [23].

Regions	Canada	Chile	Norway	Scotland
**Canada**	0.00000			
**Chile**	0.25198	0.00000		
**Norway**	0.12585	0.08865	0.00000	
**Scotland**	0.14977	0.15440	0.04209	0.00000

**Table 5 jof-07-00713-t005:** The results of the AMOVA of *Saprolegnia* spp. isolates using 163 AFLP markers; d.f., degree of freedom.

Regions	d.f.	Sum of Squares	Variance Components	% of Variation	*F*_ST_ Value
Among countries	3	101.763	1.86936	8.99	0.089
Within countries	30	567.884	18.92947	91.01	
Total		669.647	669.647		

## Data Availability

Data supporting the reported results can be found at https://www.vetinst.no/en.

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
