# Peer review of "Genetic Analyses of Saprolegnia Strains Isolated from Salmonid Fish of Different Geographic Origin Document the Connection between Pathogenicity and Molecular Diversity"

_jof, 2021, doi:10.3390/jof7090713_

Round 1

Reviewer 1 Report

Overall a really nice piece of research.

There is still the need to find better/easier/quicker method(s) to identify at species level different Saprolegnia isolates. We are all aware that some of the sequences in GenBank are outdated in the sense that some species' names have been resolved and changed in the past 10 years. This may mask some of the results of the most up-to-date techniques.

Personally, I would like to see one of the AFLP gels.

Some minor corrections are needed:

Line 59-61: please revise your sentence

Lines 67/69-71/154: please check that all species names are in italic

Table 1: repeated N isolates at the bottom of the table

Fig. 3 (2): check the sequence of the figures as there are 2 fig.3.  This figure legend needs a bit more description.

I think it would be beneficial to add on fig. 3 (1) some of the isolates characteristics: pathogenicity, hooks, etc. It would make the figure self-explanatory.

In the manuscript in vivo should be in italic.

Line 295: check the citation, as the author's name is misspelled.

Author Response

We would like to thank you and the reviewers for your useful comments, which helped us to improve our manuscript (Manuscript ID: jof-1355968 ‘Genetic analyses of Saprolegnia strains isolated from salmonid fish of different geographic origin documents connection be-tween pathogenicity and molecular diversity’).

Below, we explain in detail how we have dealt with the comments made by the reviewers.

As you will see, all but one of the reviewers’ comments have been implemented in the revised manuscript. We submit two manuscript versions, one with all changes highlighted by tracked changes and one cleaned-up version.

We hope that you will find the revised manuscript acceptable for publication and we are looking forward to your reply.

On behalf of the authors yours sincerely,

Dr. Abdelhameed Elameen

Based on the comments from the reviewers (In Italics), we have done the following revisions – our answers are given in standard font and each answer is numbered:

Review Report 1

Open Review

English language and style

( ) Extensive editing of English language and style required
( ) Moderate English changes required
(x) English language and style are fine/minor spell check required
( ) I don't feel qualified to judge about the English language and style

Yes

Can be improved

Must be improved

Not applicable

Does the introduction provide sufficient background and include all relevant references?

(x)

( )

( )

( )

Is the research design appropriate?

(x)

( )

( )

( )

Are the methods adequately described?

(x)

( )

( )

( )

Are the results clearly presented?

( )

(x)

( )

( )

Are the conclusions supported by the results?

(x)

( )

( )

( )

Comments and Suggestions for Authors

Overall a really nice piece of research.

There is still the need to find better/easier/quicker method(s) to identify at species level different Saprolegnia isolates. We are all aware that some of the sequences in GenBank are outdated in the sense that some species' names have been resolved and changed in the past 10 years. This may mask some of the results of the most up-to-date techniques.

Personally, I would like to see one of the AFLP gels.

Authors answer #1

Unfortunately, there are some logistic issues regarding the AFLP gel pictures. The Norwegian Veterinary Institute is currently moving to new locations. We sincerely regret that we did not make digitalized backups, as the original AFLP gel pictures for the time being are placed somewhere in large containers, and we have consequently not been able to provide them.

Some minor corrections are needed:

Line 59-61: please revise your sentence

Authors answer #2

We have revised the sentence in the materials and methods according to the reviewer’s suggestion. It reads now: Thirsty seven Saprolegnia spp. strains were collected from salmonids in Canada (5), Chile (6), Norway (14), Scotland (9) and Japan (3) are investigated in the study (Table 1).

Lines 67/69-71/154: please check that all species names are in italic

Authors answer #3

Changed according to the reviewer’s suggestion.

Table 1: repeated N isolates at the bottom of the table

Authors answer #4

The repeated N isolates in at the bottom of table 1, were eliminated.

Fig. 3 (2): check the sequence of the figures as there are 2 fig.3.  This figure legend needs a bit more description.

Authors answer #5

We corrected the sequence of the figures and fig 3(2) is now read as fig 4. The legend of the figure is more described and read: Figure 4. Structure analyses summary. A is bar plots showing the relationships for the all 37 Saprolegnia strains in the study assessed based by AFLP data and each strain is represented by a vertical line. The Saprolegnia isolates are sorted according to Id number as in table1. B is the values for 7 independent runs for each of K=1-7 and DeltaK = mean (L’’(K)I)/s(L(K)).

I think it would be beneficial to add on fig. 3 (1) some of the isolates characteristics: pathogenicity, hooks, etc. It would make the figure self-explanatory.

Authors answer #6

We do agree with reviewers, but when we added the information of the isolates characteristics on the figure, resulted in large text in the figure and it became difficult to read, thus we left unchanged and we referred to table 1 and now reads: Figure 3. Phylogram of isolates of Saprolegnia spp as revealed by sequencing of ITS region 2. The isolates are described by their geographic origin, ability to produce long pines on cysts and pathogenicity to salmon in table 1.

In the manuscript in vivo should be in italic.

Authors answer #7

Changed according to the reviewer’s suggestion.

Line 295: check the citation, as the author's name is misspelled.

Authors answer #8

The author’s names were corrected and it was in line 395.

Reviewer 2 Report

jof-1355968- “Genetic analyses of Saprolegnia strains isolated from salmonid fish of different geographic origin documents connection between pathogenicity and molecular diversity”.

GENERAL COMMENT:

The work entitled “Genetic analyses of Saprolegnia strains isolated from salmonid fish of different geographic origin documents connection between pathogenicity and molecular diversity” is a good work.

This study aimed to study the genetic diversity and relationships of Saprolegnia spp. (one of the most important Oomycetes pest on salmon and trout species) collected from Canada, Chile, Japan, Norway and Scotland. For this purpose, the amplified fragment length polymorphism (AFLP) and sequence data of the internal transcribed spacer (ITS) methods were used

The subject of the study is original, interesting, and topical.

Central argument is supported by evidence and analysis.

The methodology described by the author is very accurate.

This work is a good work, it only needs some minor changes, for this reason I require minor revision.

DETAILED COMMENT:

  • Title

-The title is adequate.

  • Abstract

-The abstract is well structured, and the objective of the study is clearly described.

Keywords are adequate

  • Introduction

The introduction section is not exhaustive and needs to be expanded. I suggest citing other previous works on the subject.

  • Materials and Methods

The section is well written and accurate.

  • Results

This section is accurate and detailed

  • Discussion

The discussion section is exhaustive and adequately discussed.

  • Tables and figures

Tables and Figures are clear and understandable.

  • References

The references are adequate.

Author Response

We would like to thank you and the reviewers for your useful comments, which helped us to improve our manuscript (Manuscript ID: jof-1355968 ‘Genetic analyses of Saprolegnia strains isolated from salmonid fish of different geographic origin documents connection be-tween pathogenicity and molecular diversity’).

Below, we explain in detail how we have dealt with the comments made by the reviewers.

As you will see, all of the reviewers’ comments have been implemented in the revised manuscript. We submit two manuscript versions, one with all changes highlighted by tracked changes and one cleaned-up version.

We hope that you will find the revised manuscript acceptable for publication and we are looking forward to your reply.

On behalf of the authors yours sincerely,

Dr. Abdelhameed Elameen

Based on the comments from the reviewers (In Italics), we have done the following revisions – our answers are given in standard font and each answer is numbered:

Review Report 2

Open Review

English language and style

( ) Extensive editing of English language and style required
( ) Moderate English changes required
(x) English language and style are fine/minor spell check required
( ) I don't feel qualified to judge about the English language and style

Yes

Can be improved

Must be improved

Not applicable

Does the introduction provide sufficient background and include all relevant references?

(x)

( )

( )

( )

Is the research design appropriate?

(x)

( )

( )

( )

Are the methods adequately described?

(x)

( )

( )

( )

Are the results clearly presented?

(x)

( )

( )

( )

Are the conclusions supported by the results?

(x)

( )

( )

( )

Comments and Suggestions for Authors

jof-1355968- “Genetic analyses of Saprolegnia strains isolated from salmonid fish of different geographic origin documents connection between pathogenicity and molecular diversity”.

GENERAL COMMENT:

The work entitled “Genetic analyses of Saprolegnia strains isolated from salmonid fish of different geographic origin documents connection between pathogenicity and molecular diversity” is a good work.

This study aimed to study the genetic diversity and relationships of Saprolegnia spp. (one of the most important Oomycetes pest on salmon and trout species) collected from Canada, Chile, Japan, Norway and Scotland. For this purpose, the amplified fragment length polymorphism (AFLP) and sequence data of the internal transcribed spacer (ITS) methods were used

The subject of the study is original, interesting, and topical.

Central argument is supported by evidence and analysis.

The methodology described by the author is very accurate.

This work is a good work, it only needs some minor changes, for this reason I require minor revision.

DETAILED COMMENT:

  • Title

-The title is adequate.

  • Abstract

-The abstract is well structured, and the objective of the study is clearly described.

Keywords are adequate

  • Introduction

The introduction section is not exhaustive and needs to be expanded. I suggest citing other previous works on the subject.

Authors answer #1

According to the suggestion by the reviewer, we have included and citied other previous works on the subject in the section of the introduction.

  • Materials and Methods

The section is well written and accurate.

  • Results

This section is accurate and detailed

  • Discussion

The discussion section is exhaustive and adequately discussed.

  • Tables and figures

Tables and Figures are clear and understandable.

  • References

The references are adequate.